# Analysis of the UDP-Glucosyltransferase (*UGT*) Gene Family and Its Functional Involvement in Drought and Salt Stress Tolerance in *Phoebe bournei*

**DOI:** 10.3390/plants13050722

**Published:** 2024-03-04

**Authors:** Hengfeng Guan, Yanzi Zhang, Jingshu Li, Zhening Zhu, Jiarui Chang, Almas Bakari, Shipin Chen, Kehui Zheng, Shijiang Cao

**Affiliations:** 1College of Forestry, Fujian Agriculture and Forestry University, Fuzhou 350002, China; g1290570831@163.com (H.G.); chuchu7613@163.com (J.L.); zheningzhu@163.com (Z.Z.); cherichang@163.com (J.C.); almasbakari154@gmail.com (A.B.); fjcsp@126.com (S.C.); 2Center for Plant Metabolomics, Haixia Institute of Science and Technology, Fujian Agriculture and Forestry University, Fuzhou 350002, China; zhangyanzi163@163.com; 3College of Computer and Information Sciences, Fujian Agriculture and Forestry University, Fuzhou 350002, China

**Keywords:** bioinformatics analysis, *Phoebe bournei*, UDP-glycosyltransferase, expression analysis

## Abstract

Uridine diphosphate glycosyltransferases (UDP-GTs, UGTs), which are regulated by *UGT* genes, play a crucial role in glycosylation. In vivo, the activity of *UGT* genes can affect the availability of metabolites and the rate at which they can be eliminated from the body. *UGT* genes can exert their regulatory effects through mechanisms such as post-transcriptional modification, substrate subtype specificity, and drug interactions. *Phoebe bournei* is an economically significant tree species that is endemic to southern China. Despite extensive studies on the *UGT* gene family in various species, a comprehensive investigation of the *UGT* family in *P*. *bournei* has not been reported. Therefore, we conducted a systematic analysis to identify 156 *UGT* genes within the entire *P*. *bournei* genome, all of which contained the PSPG box. The *PbUGT* family consists of 14 subfamilies, consistent with *Arabidopsis thaliana*. We observed varying expression levels of *PbUGT* genes across different tissues in *P*. *bournei*, with the following average expression hierarchy: leaf > stem xylem > stem bark > root xylem > root bark. Covariance analysis revealed stronger covariance between *P*. *bournei* and closely related species. In addition, we stressed the seedlings with 10% NaCl and 10% PEG-6000. The *PbUGT* genes exhibited differential expression under drought and salt stresses, with specific expression patterns observed under each stress condition. Our findings shed light on the transcriptional response of *PbUGT* factors to drought and salt stresses, thereby establishing a foundation for future investigations into the role of *PbUGT* transcription factors.

## 1. Introduction

Glycosylation, a widespread compound modification pathway in plant secondary metabolism, is a crucial transformation process in organisms. It serves as the final stage for many biosynthetic products, altering the activity, stability, and solubility of receptors [1]. Glycosylation plays a vital role in plant hormone balance, detoxification processes, and secondary metabolite production. Glycosyltransferases (GTs) are responsible for plant glycosylation and constitute a highly diverse multi-gene family present in various organisms. These enzymes catalyze the specific transfer of glycosyl groups from donors to receptors, forming α- and β-glycosidic bonds and generating oligosaccharides, polysaccharides, complex sugars (such as glycoproteins and glycolipids), and glycoside compounds [2]. The most prevalent glycosyl donor in plants is UDP-glucose, while receptors include glycolipids, proteins, and nucleic acids.

The carbohydrate active enzyme database (CAZy) classifies glycosyltransferases (GTs) from different species into 114 gene families based on amino acid sequence similarity, substrate specificity, catalytic mechanism, signal membrane, and conformation of the glycosidic chain. The GT1 branch represents the largest division, indicating its close functional association with plants. Within the GT1 branch, there exists a subclass known as UDP-glucosyltransferases (UGTs) due to their utilization of UDPG as the glycosyl donor [3]. One prominent characteristic of the plant *UGT* family is the presence of a highly conserved segment, known as the PSPG box, comprising 44 amino acids located at the protein’s C-terminus [4]. This conserved motif facilitates the recognition of and interaction with uridine diphosphate glycosyl donors. Conversely, the N-terminal sequence, which is considerably variable and less conserved, governs the recognition and binding of various receptors, contributing to the vast substrate diversity and complex structures of plant *UGT* genes, such as their involvement in the glycosylation of diverse compounds including hormones, terpenoids, and sterols [5].

The *UGT* gene family has been extensively identified and classified in various plants, including *Arabidopsis thaliana* [6], *Linum usitatissimum* [7], *Triticum aestivum* [8], *Zea mays* [9], *Gossypium hirsutum* [10], and *Prunus persica* [11]. *Arabidopsis thaliana*, a model plant, possesses 122 *UGT* genes, among which 8 are pseudogenes. These genes are categorized into 14 distinct groups (A–N) based on their amino acid sequences [6]. Subsequent studies have revealed the presence of O and P groups in higher plants [12]. *Oryza sativa* has a total of 609 *UGT* genes, divided into 40 groups along with an unknown class [13]. *Citrus grandis* exhibits 145 *UGT* genes distributed among 16 groups [14], while *Zea mays* contains 147 *UGT* genes classified into 17 groups, with 18 genes shared with *Arabidopsis thaliana* and 2 genes shared with *Oryza sativa* [9]. The functional roles of the *UGT* gene family have been validated in various plants, including hormone balance regulation, localization control, secondary metabolite synthesis, detoxification, and defense responses [15].

*Phoebe bournei*, a species of the Lauraceae family, is characterized by a straight trunk and limited branches. It serves as an excellent timber species and is endemic to China [16]. Noteworthy for its unique fragrance, *P*. *bournei* holds significant value as a timber tree. Its wood exhibits strong resistance to corrosion, making it resistant to cracking and bending during the drying process. Consequently, it possesses high economic worth and finds applications in shipbuilding and furniture production [17]. However, due to excessive logging driven by its economic value and the adverse effects of environmental factors such as soil fertility degradation, *P*. *bournei* has experienced a decline in its wild population and is now listed as a secondary nationally protected plant.

While several gene families in other plants have been extensively studied, limited research has been conducted on the gene family of *P*. *bournei*. This dearth of studies can be attributed to the relatively recent sequencing of the *P*. *bournei* genome. The complete genome data of *P*. *bournei* was initially obtained by Liu Zhongjian and Chen Shiping from Fujian Agriculture and Forestry University in 2020 [18]. However, this study did not unveil the chromosomal-level information. Subsequently, in 2022, the genome of *P*. *bournei* was deciphered by the Tong Zaikang team from Zhejiang Agriculture and Forestry University, shedding light on the synthesis mechanism of *P*. *bournei* sesquiterpenes [19].

The gene families in *P*. *bournei* primarily include commonly found families like *WRKY* [20], *NF-Y* [21], and *PLR* [22], while the *UGT* gene family has received limited attention. Previous studies have primarily focused on whole-gene family analyses, with few investigations on stress responses. Furthermore, existing research primarily addresses nutrient deficiency stress, neglecting drought and salt stresses. Therefore, our study aims to elucidate the response mechanisms of *PbUGT* genes under drought and salt stress conditions.

Previously, we studied transcriptional analysis of the *GATA* gene family under abiotic stress in Phoebe bournei [23]. We know that under drought stress, the photosynthesis of *P. fortunei* is inhibited, the net photosynthetic rate, transpiration rate, and stomatal conductance of its leaves decrease, while the activities of oxidoreductase (POD and SOD) increase, and the ABA content in its roots increases. In terms of morphological characteristics, the root color of the seedlings becomes darker with longer drought treatment [24]. Moreover, in our previous experiments, we found that excessive stress time and excessive solution concentration may lead to the death of the seedlings. Therefore, we will use the method used in these previous experiments to stress *Phoebe bournei*, using 10% NaCl to apply salt stress to annual seedlings and 10% PEG-6000 to apply drought stress. Finally, qRT-PCR was used to detect the expression profile of the genes under drought and salt stress. This study provides a new idea for studying the evolution of *UGT* genes and helps to provide a better understanding of the abiotic stress response of *Phoebe bournei*.

## 2. Results

### 2.1. Identification and Physicochemical Analysis of PbUGT Gene Members

Through rigorous verification using various methods, a comprehensive set of 156 *PbUGT* genes was successfully identified within the entirety of the genome of *P*. *bournei*. These *UGT* genes were subsequently designated *PbUGT*1-156 based on their chromosomal distribution and are presented in Table 1. To gain further insights into the physicochemical properties of the *UGT* proteins, ProtParam, MEME, and other online tools were employed for analysis. The results revealed that the amino acid count of the 156 *PbUGT* proteins ranged from 132 aa (*PbUGT*66) to 1019 aa (*PbUGT*89). The molecular weight of the proteins varied between 14948.04 Da (*PbUGT*66) and 113398.82 Da (*PbUGT*89). The isoelectric points spanned a range from 4.62 to 9.80. Specifically, there were 136 acidic proteins (PI < 6.5), 12 neutral proteins (6.5 < PI < 7.5), and 8 basic proteins (PI > 7.5). Subcellular localization analysis of these *PbUGT* proteins indicated that 88 proteins were localized within the chloroplast, 43 were in the cytoplasm, and 15 were in the nucleus, while the remaining proteins were distributed across various organelles such as vacuoles, peroxisomes, and mitochondria. Furthermore, instability coefficient analysis unveiled that 46 proteins were classified as stable, whereas the remaining 110 proteins exhibited instability. Notably, the maximum instability coefficient reached a significant value of 73.59, suggesting that the majority of the *PbUGT* proteins were characterized as unstable acidic proteins.

### 2.2. Chromosome Localization Analysis of the PbUGT Gene Family

To determine the precise chromosomal positions of the *UGT* genes, we utilized MapChart 2.32 software to construct a visual representation of their distribution. Our analysis revealed that the 156 *UGT* genes were dispersed across all 12 chromosomes. Notably, chromosome 10 harbored the highest number of *PbUGT*s, with a total of 32 genes, followed by chromosome 2, with 28 *PbUGT*s. Furthermore, the distribution pattern of *UGT* gene members within different subfamilies of *P*. *bournei* displayed irregularities, characterized by the presence of gene clusters. These clusters varied in size, ranging from 2 to 13 genes, with the largest cluster residing on chromosome 10. We further examined the chromosomal distribution within different gene groups. Chromosome 2 contained 28 genes belonging to eight distinct groups. Among these, group A exhibited the greatest number of members in the *P*. *bournei UGT* family, totaling 36 genes. Specifically, thirteen members from this group formed a cluster on chromosome 7, while four members clustered on chromosomes 2 and 4. The remaining six members were dispersed across chromosomes 1, 2, 5, and other regions encompassing chromosomes 8 to 12 (Figure 1). This uneven distribution of *PbUGT* genes across chromosomes indicates the presence of genetic variations during the evolution of *P*. *bournei*.

### 2.3. PbUGT Protein Conserved Motif Analysis

To gain further insights into the conserved domain characteristics of the *UGT* family in *Phoebe bournei*, we employed the online tool MEME to identify and assign 10 distinct motifs, designated as motifs 1 to 10 (Figure 2). The motif analysis revealed that the number of motifs present in each *PbUGT* family member ranged from three to thirteen. Notably, 114 out of the 156 members exhibited a combination of motifs 4 to 9. This unique motif combination serves as a distinctive feature of the family, setting it apart from other gene families and implying its close association with specific functional and performance sequences. Additionally, a notable observation was that members belonging to the same evolutionary group exhibited a high degree of similarity in the distribution of conserved motifs. This suggests that members classified within the same group are likely to share similar functions due to the similarity in motif distribution. Domain analysis unveiled the presence of four conserved domains, namely glycosyltransferase, GTB-type superfamily, GT1-Gtf-like, and PLN02448 domains, in all 156 *UGT* members. This observation suggests that the composition of these three conserved motifs is intricately linked to the synthesis and regulation of glycosyltransferases.

### 2.4. Phylogenetic Analysis of the PbUGT Gene Family

Previous studies have classified *UGT* genes in *Arabidopsis* into 14 distinct groups, labeled A to N, with the identification of three additional groups, O, P, and Q, in higher plants. To elucidate the functional attributes of each *PbUGT*-family gene and determine its phylogenetic group, we obtained 122 sequences from *A*. *thaliana* via the *Arabidopsis* resource website. Subsequently, we constructed a phylogenetic tree by incorporating these sequences with the 156 identified *P*. *bournei* sequences. The phylogenetic tree revealed the presence of 14 *A*. *thaliana* groups within *P*. *bournei*, with an uneven distribution across these groups (Figure 3). Notably, the majority of *PbUGT* genes were found in groups A, G, E, and L, with the largest representation in group A, consisting of 36 members. Conversely, groups K, M, and N each contained only one member. The differences observed among various subfamilies may be linked to the evolutionary trajectory and functional diversity within *P*. *bournei*.

### 2.5. Organ and Tissue Expression Profile Analysis of the PbUGT Gene Family

By leveraging the original transcriptome sequencing data obtained from five distinct tissues in *P*. *bournei* (PRJNA628065), we constructed an expression heatmap to examine the tissue-specific expression patterns of the 156 *PbUGT* genes (Figure 4). The heatmap displayed expression levels ranging from low (blue) to high (dark red), highlighting considerable variations in the expression patterns across different tissues. Among the genes analyzed, *PbUGT*1, *PbUGT*21, *PbUGT*26, *PbUGT*27, *PbUGT*35, *PbUGT*55, *PbUGT*84, *PbUGT*104, *PbUGT*117, *PbUGT*121, *PbUGT*145, and *PbUGT*154 lacked corresponding expression data in the reported transcriptome, suggesting their limited involvement in the growth of *P*. *bournei*. In contrast, *PbUGT*101 and *PbUGT*136 exhibited significantly higher expression levels (FPKM > 15) in all tissues, indicating their crucial roles in the growth and development of *P*. *bournei*. On average, the expression levels ranked as follows: leaf > stem xylem > stem epidermis > root xylem > root epidermis. Among the fourteen subfamilies, the B subfamily displayed the highest average expression level, surpassing other subfamilies by a substantial margin. Conversely, the C, D, and K subfamilies exhibited notably low overall expression levels (FPKM < 1). The differential expression of *PbUGT* genes across various tissues signifies functional disparities among genes within distinct subfamilies.

### 2.6. Synteny Analysis of Homologous UGT Genes among Phoebe bournei, Arabidopsis thaliana, Populus trichocarpa, and Cinnamomum camphora

To investigate the collinearity relationships within the *UGT* gene family among *Phoebe bournei*, *Arabidopsis thaliana*, *Populus trichocarpa*, and *Cinnamomum camphora*, a comprehensive genome-wide collinearity analysis was conducted. The aim was to unveil the interconnections within the *UGT* family. As depicted in Figure 5, a total of 13 *PbUGT* genes on 8 chromosomes displayed collinearity with *UGT* genes from *A*. *thaliana*, while 39 *UGT* genes on 10 chromosomes exhibited collinearity with *UGT* genes from *P*. *trichocarpa*. Additionally, 60 *UGT* genes on all 12 chromosomes displayed collinearity with *UGT* genes from *Cinnamomum camphora*. The collinearity analysis revealed that species with closer genetic relationships exhibited stronger collinearity within the same gene family. This observation suggests a plausible association with species evolution, where these genes might have arisen earlier, with each gene family demonstrating homology prior to species differentiation. Over time, as the species diverged, these homologous gene relationships gradually diminished.

### 2.7. Duplication Analysis of PbUGT Genes in Chromosomal Replication

To elucidate the expansion and evolutionary mechanisms underlying the *PbUGT* gene family, potential gene duplication events in the *P*. *bournei* genome were investigated. The entire *P*. *bournei* genome was compared using MCScanX software, leveraging amino acid sequence homology. The analysis revealed the presence of 14 tandem repeat gene clusters within the *UGT* gene family members (Figure 6), indicating that tandem repeats played a pivotal role in the gene family’s expansion. Additionally, this study examined the ratio of nonsynonymous substitutions (Ka) to synonymous substitutions (Ks) between the duplicated genes. The results demonstrated that the ratio was significantly below one, implying a negative selection effect on the *PbUGT* gene during the evolutionary process, indicative of purifying selection.

### 2.8. Enrichment Analysis of Stress-, Hormone-, and Light-Responsive Cis-Elements in the Promoter Region of Phoebe bournei UGT Genes

Cis-acting elements serve as indicators of gene function and predictors of transcriptional regulation patterns. The promoter regions of *PbUGT* genes encompass numerous cis-acting elements with distinct functions, which can be categorized into the stress response, hormone response, and light response. In total, 7, 35, and 11 cis-elements associated with stress responsiveness, hormone responsiveness, and light responsiveness, respectively, were identified across all *PbUGT* promoter sequences (Figure 7). Among them, the G-box (a light-response cis-element) exhibited the highest frequency, occurring 366 times in the promoter sequences of 131 genes. It was followed by ABRE (a hormone-response cis-element), which appeared 326 times in the promoter sequences of 124 genes. Previous studies have indicated that cis-element responses in various plants are involved in diverse stresses and hormone regulation. Hence, *PbUGT* genes may play a role in stress, hormone, and light regulation during the growth and development of *P*. *bournei*. This finding provides valuable insights for subsequent stress treatments in this experimental analysis.

### 2.9. Screening and Verification of Stress Resistance Genes in PbUGT Genes

Based on the cis-acting elements, five genes—*PbUGT*14, *PbUGT*15, *PbUGT*56, *PbUGT*60, and *PbUGT*154—displaying the highest number of elements corresponding to adversity were identified. Subsequently, their transient expression levels were verified in response to various adversity stresses using RT-qPCR. The expression level analysis of *PbUGT* genes under 10% NaCl salt stress revealed that *PbUGT*14, *PbUGT*15, *PbUGT*56, *PbUGT*60, and *PbUGT*154 were all affected by salt stress (Figure 8A). Generally, their expression levels exhibited a “rise–fall–rise” trend over time. *PbUGT*14, *PbUGT*56, *PbUGT*60, and *PbUGT*154 reached their peak expression levels at 4 h, followed by an upward trend after a sudden decline. In contrast, the expression level of *PbUGT*15 continuously decreased with time.

Under 10% PEG-6000 drought stress (Figure 8B), except for *PbUGT*15, *PbUGT*14, *PbUGT*56, *PbUGT*60, and *PbUGT*154 displayed peak expression levels at 6 h, followed by a sharp decrease and subsequent upward trend. *PbUGT*15, on the other hand, peaked at 4 h, exhibiting an overall “rise–decline–rise” pattern. The expression levels of the five genes demonstrated significant variations under 10% NaCl salt stress and 10% PEG-6000 drought stress, displaying both up-regulation and down-regulation. This indicates that these members of the gene family play crucial roles in the response of *P*. *bournei* to drought and salt stresses.

## 3. Discussion

In the present investigation, a total of 156 *PbUGT* genes were identified in the genome of *Phoebe bournei*. The number of *PbUGT* genes in *P*. *bournei* exceeded that of *Cucumis sativus L*. (94), *Dendrobium officinale* (44), and *pomegranate* (145), but was lower than *Arachis hypogaea L*. (267) and *Glycine max* (182). In comparison to other gene families, the *UGT* gene family demonstrated a higher abundance across all plant species. The initial stage of species differentiation showed a certain covariance, and the analysis of species covariance showed that *UGT* genes may have differentiated earlier in the evolutionary process, showing higher homology and closer linkage relationship with related plants. As species continue to differentiate, covariance gradually decreases.

Analysis of the physicochemical properties of *P*. *bournei* identified that 145 genes (93%) had an average molecular mass greater than 30 KDa, with an average of 50 KDa, and contained more than 100 amino acids. Regarding the proteins, 136 (87%) were found to be acidic, and 130 (83%) were hydrophilic. The instability coefficients of 110 proteins exceeded 40, indicating that most of the proteins regulated by *PbUGT* genes were both unstable and acidic. Subcellular localization analysis revealed an uneven distribution of *PbUGT* genes across organelles, with the majority located in the mitochondria (88), followed by 43 in the cytoplasm, and 15 in the nucleus, with the remaining in organelles such as vesicles and peroxisomes. The abundance of *PbUGT* genes in mitochondria may be related to the substantial energy requirements of glycosyltransferase, a process regulated by *UGT*. This finding provides a significant foundation and insight for further *UGT* gene research.

During natural growth, forest trees frequently encounter various abiotic stresses, including drought, low temperature, and salinity stresses, which impede their growth. These stresses can cause damage to plants, necessitating optimal water and fertilizer conditions for forest trees to achieve expected growth rates. However, this also results in greater nutrient consumption and longer recovery periods for the forest land. Genes play a vital role in enhancing the tolerance of forest trees to abiotic stresses [25]. Forest trees respond to stress signals by modulating the expression of various genes, which, in turn, regulate metabolic responses and control tree growth under stressful conditions.

The current experiment, in conjunction with previous studies and cis-element analysis, has demonstrated that *UGT* genes across species respond to diverse abiotic stresses. For instance, maize mutants lacking the *ZmUGT2* gene failed to complete flavonol glycosylation in vivo, resulting in inhibited germination rates and suppressed stem and root growth under abiotic stress conditions, particularly salt and drought stresses [26,27]. In tea trees, abiotic stress suppressed the gene regulation of *CsUGT91Q2* responsible for nerolidol glycol production, leading to impaired reactive oxygen species scavenging and reduced cold resistance, thereby increasing sensitivity to cold stress [28,29].

As transcription factors, *UGT* genes can activate the expression of stress-related genes, enhancing plant stress resistance and participating in the regulatory network governing adversity stress in forest trees. Phylogenetic analysis, transcriptome data, and RT-qPCR data confirmed that the five *PbUGT* genes can be rapidly induced and activated in response to abiotic stresses such as drought and salt stress, playing a significant role in the plant’s response to these stresses.

Furthermore, transcriptome data analysis of *P*. *bournei* seedlings subjected to drought and salt stress revealed the involvement of all five *UGT* genes in abiotic stress responses. Their expression varied significantly with the duration of stress. The majority of genes exhibited up-regulated expression under drought and salt stress, displaying a “rising–declining–rising” expression pattern under both 10% NaCl salt stress and 10% PEG-6000 drought stress. Particularly, under the 10% NaCl salt stress, all five *UGT* genes exhibited a high expression pattern after only 4 h. These findings suggest that *PbUGT* genes play a positive regulatory role in the abiotic stress response. This provides new insights for *P*. *bournei* species selection and stress regulation studies, aiming to expand the planting area of *P*. *bournei* and maximize its growth potential under stressful conditions.

## 4. Materials and Methods

### 4.1. Identification, Chromosomal Localization, and Analysis of Protein Physicochemical Properties

The whole-genome data and annotation files of *Phoebe bournei* were acquired from the China National GeneBank DataBase (CNGBdb) with lookup number CNP0002030 (https://db.cngb.org/ (accessed on 1 June 2023)). The Pfam website was utilized to obtain the *UGT* structural domain (query code PF00201) for screening conserved sequences (E-value < 1 × 10^−5^) in all *P*. *bournei* protein sequences using Hmmer 3.3.2 software (http://hmmer.org/ (accessed on 2 June 2023)) [30]. Furthermore, the CDD function of the NCBI website (https://www.ncbi.nlm.nih.gov/cdd (accessed on 3 June 2023)) and SMART online (https://smart.embl.de/ (accessed on 4 June 2023)) [31] were employed to analyze and integrate the obtained protein sequences. Erroneous sequences were manually removed, ensuring the presence of UDP-glycosyltransferase structural domains. Ultimately, 156 sequences containing the complete *UGT* structural domain were selected.

Based on the whole-genome data and annotation files of *P*. *bournei*, specific location information for *UGT* genes on each chromosome was obtained. Subsequently, Mapchart 2.32 software [32] was employed to map the location information of each *PbUGT* gene onto the 12 *P*. *bournei* chromosomes. The ExPASy online (https://web.expasy.org/protparam (accessed on 15 July 2023)) [33] was utilized to predict and calculate the molecular weight, number of amino acids, theoretical PI (isoelectric point), number of positive (negative) residues, instability coefficient, lipolysis coefficient, and total average hydrophilicity of each *UGT* protein in *P*. *bournei*. The subcellular localization of each member of the *P*. *bournei UGT* gene family was predicted using the WoLFPSORT website (https://wolfpsort.hgc.jp (accessed on 17 July 2023)) [34].

### 4.2. Gene Structure, Protein Conserved Motifs, and Exon–Intron Analysis

The annotation file of the *P*. *bournei* genome and the amino acid sequences of the 156 *UGT* genes were used to construct a phylogenetic evolutionary tree for *PbUGT* genes using the One Step Build an ML Tree function in TBtools v1.120 [35]. The MEME 5.5.2 online software (http://meme-suite.org/tools/meme (accessed on 19 July 2023)) [36] was employed to analyze the conserved motifs of *PbUGT* genes with a motif set of 10. The GSDS2.0 online tool (http://gsds.cbi.pku.edu.cn/ (accessed on 21 July 2023)) [37] was utilized to map the gene structure, including the UTRs, exons, and introns of each *UGT* gene. The Batch CD Search function of the NCBI website (https://www.ncbi.nlm.nih.gov/Structure/bwrpsb/bwrpsb.cgi (accessed on 23 July 2023)) was used to query the structural domains by submitting the gene sequences. The Gene Structure View function in TBtools [35] facilitated the visualization and mapping of phylogenetic relationships, conserved motif composition, structural domains, and exon–intron gene structures for each *PbUGT* gene.

### 4.3. Phylogenetic Analysis and Construction of the Phylogenetic Tree

Nucleotide and amino acid sequences of 122 *Arabidopsis thaliana UGT* genes were obtained from the *A. thaliana* glycosyltransferase family 1 information website (http://www.p450.kvl.dk/UGT.shtml (accessed on 27 July 2023)) [38,39,40] as a reference. Multi-protein sequence alignment was performed using MUSCLE [41] on the combined sequences of 156 *UGT* genes from *P*. *bournei*. Phylogenetic trees of related proteins were constructed using the maximum likelihood method in the MEGA11 program with a bootstrap value set to 1000 and default parameters [42]. The constructed phylogenetic trees were further organized and embellished using the iTol 6.8.2 online website (https://itol.embl.de/ (accessed on 30 July 2023)) [43].

### 4.4. Analysis of Tissue Expression Profiles

The sequence number PRJNA628065 of *P*. *bournei* was searched on the European Bioinformatics Centre repository EBI (https://www.ebi.ac.uk (accessed on 1 October 2023)). Transcriptome expression data of *P*. *bournei* in five different tissues were downloaded [19]. RNA-seq analysis was conducted on the data to investigate the differential expression of the 156 *PbUGT* genes. Tbtools v1.120 software was used to algebraically transform the FPKM values from the analyzed data into log2 values, which were then used to generate heat maps.

### 4.5. Covariance Analysis

To elucidate the covariance patterns among *PbUGT* family homologous genes in *Arabidopsis thaliana*, *Populus hirsutus*, and *Cinnamomum camphora*, a genome-wide covariance analysis was conducted. This analysis integrated the chromosome-level genome data of *P*. *bournei* with the genome-wide data of *A*. *thaliana* [44], *P*. *hirsutus* [45], and *C*. *camphora* [46]. The Synteny Visualization function in Tbtools was employed to map the covariance across species. The whole-genome data of *A*. *thaliana* and *P*. *trichocarpa* from the Ensembl Plants database (https://plants.ensembl.org/index.html (accessed on 3 October 2023)) and *C. camphora* genome-wide data from the National Genomics Data Center (https://ngdc.cncb.ac.cn/ (accessed on 3 October 2023)) (login number: GWHBGXC00000000) were utilized.

### 4.6. Chromosomes Location, Gene Duplication, and Synteny Analysis

Circos [47] was used to identify each *PbUGT* gene on individual chromosomes. The results were then visualized using the Multiple Covariance Scanning Toolkit MCScan [48] with default settings. A comparison within the entire *P*. *bournei* genome was performed based on amino acid sequence homology. The duplication events in each *PbUGT* gene were analyzed, and the results were visualized using Circos and Dual Synteny Plot in Tbtools.

### 4.7. Enrichment Analysis of Stress-, Hormone-, and Light-Responsive Cis-Elements in the Promoter Region

The GTF/GFF3 Sequences Extract function of Tbtools was employed to obtain a 2000 bp segment upstream of the promoter region of each *PbUGT* gene based on the whole-genome sequencing sequence of *Phoebe bournei*. PlantCare (https://bioinformatics.psb.ugent.be/webtools/plantcare/html/ (accessed on 5 October 2023)) [49], an online software, was used for the prediction and screening of cis-acting elements associated with *PbUGT* genes. Subsequently, the data were categorized, organized, and analyzed, and the Heatmap function in Tbtools was utilized to classify and visualize the number of cis-acting elements for each stress-responsive *PbUGT* gene.

### 4.8. Material and Stress Treatments

One-year-old *P*. *bournei* seedlings from the Fujian Academy of Forestry were selected as the plant material and incubated in artificial climate chambers under different treatments. Before stress, the soil for planting the seedlings was prepared using peat soil, humus soil, sandy soil, perlite, etc., in a ratio of 5:2:2:1, and was cultivated in pots with a diameter of 150 mm. During stress treatment, the setting parameters of the artificial climate chambers were as follows: light cycle 12 h/d, temperature 25 °C, LED lights used for lighting, photosynthetically active radiation set to 1200 μmol·mol^−1^·s^−1^. The capacity of each pot was about 3.75 L. The seedlings with consistent growth potential were divided into two groups: the control group and the stress treatment group. The treatment group comprised thirty plants, while the control group consisted of three plants. For each biological replicate, two plants were included in the treatment group. Three sets of biological replicates were established for each time point. During stress treatment, we removed the *P*. *bournei* seedlings from the basin and cleaned their roots with pure water. The PEG treatment was transplanted into a beaker containing 10% PEG 6000 of ¼-strength Hogland solution. The salt treatment was transplanted into a beaker containing 10% NaCl of ¼-strength Hogland solution. The control group was maintained under normal moisture conditions. All stress treatments were conducted for 4, 6, 8, and 12 h. After the designated treatment periods, leaf samples from the *P*. *bournei* seedlings were collected, immediately frozen in liquid nitrogen, and stored for subsequent RNA extraction.

### 4.9. Real-Time Quantitative Polymerase Chain Reaction (RT-qPCR) Analysis of PbUGT Genes

RNA extraction was performed using the HiPure Plant RNA Mini Kit (Magenbio, Canton, China), and cDNA synthesis was carried out using the PrimeScript RT reagent Kit (Perfect Real-Time) kit (TaKaRa, Dalian, China). Based on the types of cis-acting elements present in the *PbUGT* gene family members, the genes with the highest number of elements associated with drought and salt stress responses (*PbUGT*14, *PbUGT*15, *PbUGT*56, *PbUGT*60, and *PbUGT*154) were selected for RT-qPCR analysis. The primer sequences for the quantitative RT-qPCR are shown in Table 2. Specific primers in the non-conserved regions of the target genes were designed using Primer 3.0 software and synthesized by Fuzhou Qingbaiwang Biotechnology Company. Real-time fluorescence quantitative analysis was performed using 1 μL of cDNA template, 10 μL of SYBR Premix Ex TaqTM II, 2 μL of specific primers, and 7 μL of ddH_2_O. The reaction program consisted of 40 cycles: 95 °C for 30 s, 95 °C for 5 s, 60 °C for 30 s, 95 °C for 5 s, 60 °C for 60 s, and 50 °C for 30 s. The internal reference gene used was PbEF1α (GenBank no. KX682032) [50]. The expression of the target gene was calculated using the 2^−ΔΔCt^ method. GraphPad Prism 8.0 software was utilized for statistical analysis, performing *t*-tests and generating graphs. A significance level of *p* < 0.05 was used to determine statistical significance.

## 5. Conclusions

The *UGT* gene family holds considerable significance in plant growth and development. In the case of *Phoebe bournei*, a comprehensive set of 156 *PbUGT* genes were successfully identified. Evolutionary analysis facilitated the classification of *P*. *bournei UGT* genes into fourteen distinct subfamilies. Notably, genes within the same subfamily often exhibited similar exon–intron structures, motifs, and protein configurations. The expression patterns of *PbUGT* genes demonstrated noteworthy variations under salt and drought stresses. Our experimental data substantiated the significant differences exhibited by *UGT* genes in response to diverse stresses, thereby emphasizing their crucial role in stress adaptation. This study serves as a foundational exploration into the functional attributes of the *PbUGT* gene family, offering a theoretical framework for fellow researchers interested in investigating the response of *PbUGT* genes under abiotic stress conditions. In light of the mounting challenges posed by land salinization and drought, further research on enhancing *P*. *bournei* species could contribute to expanding the cultivation area dedicated to this species. The anti-stress mechanism of *UGT* gene needs to be further explored.

## Figures and Tables

**Figure 1 plants-13-00722-f001:**
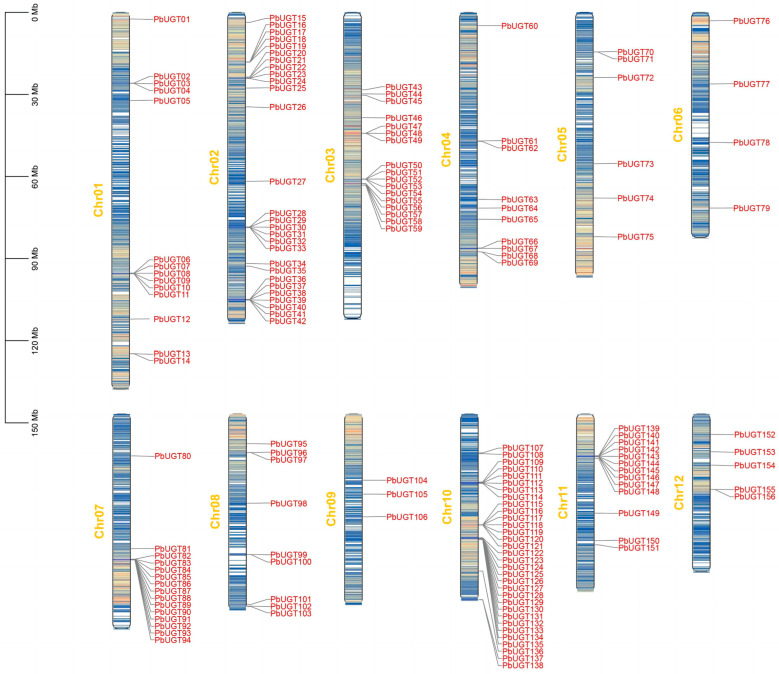
Chromosomal distribution and duplication of *PbUGT* genes.

**Figure 2 plants-13-00722-f002:**
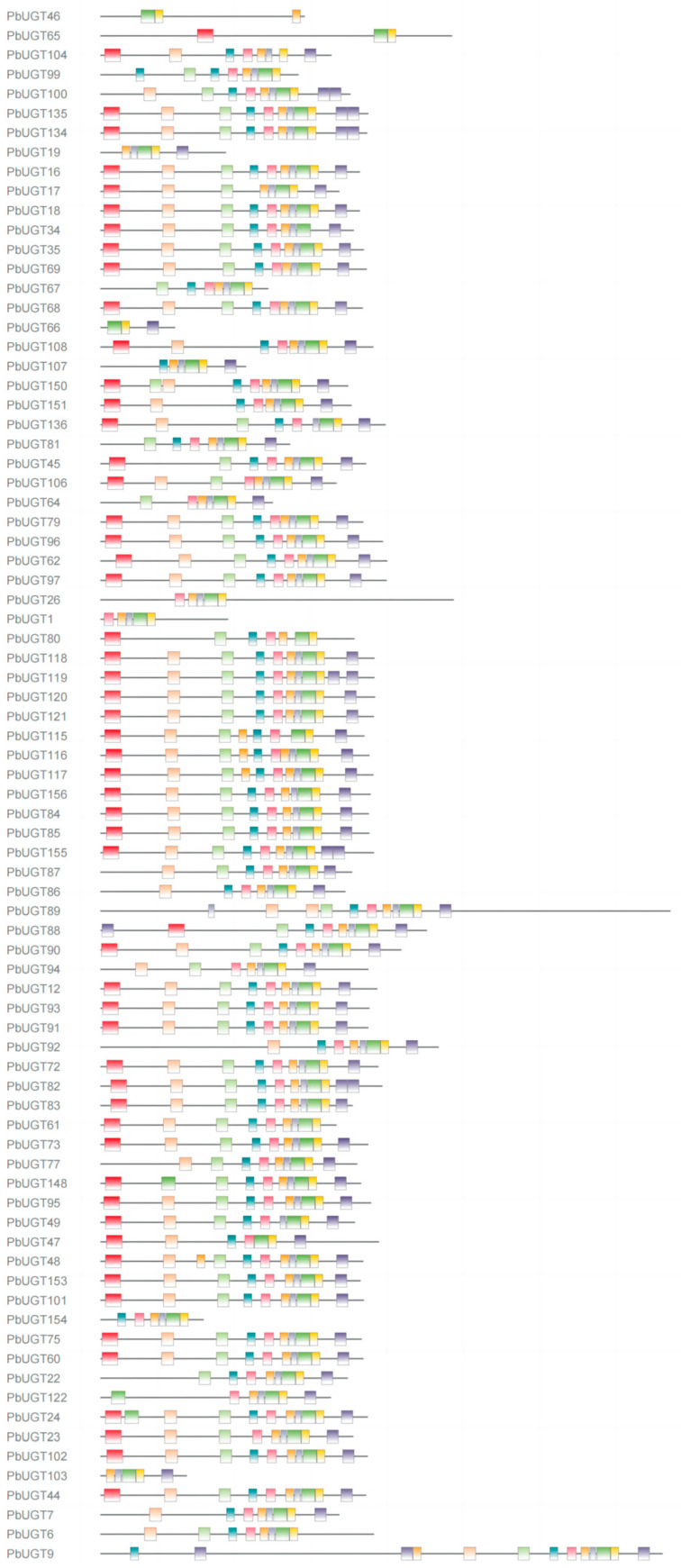
Conserved motif of the *PbUGT* gene family. Different colors correspond to different types of motifs with the numbers 1–10.

**Figure 3 plants-13-00722-f003:**
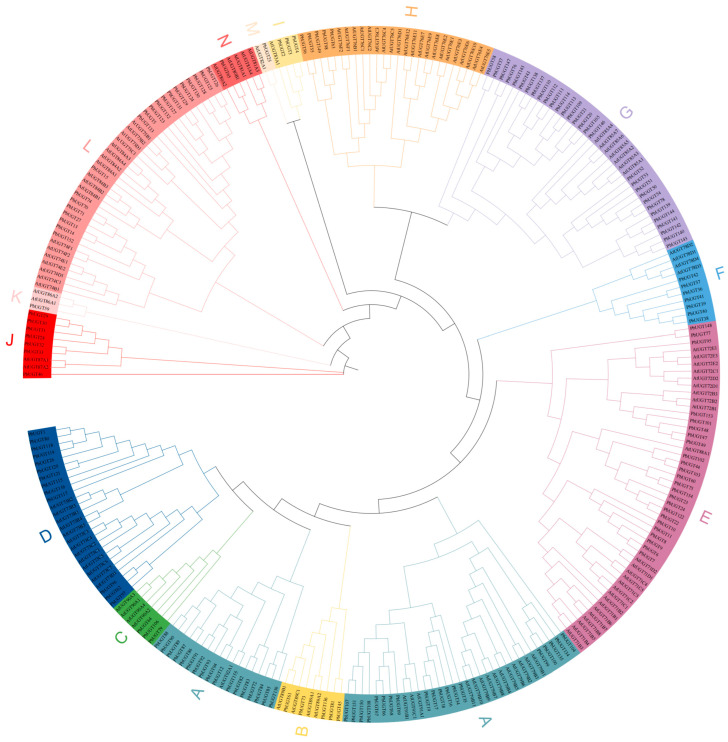
Phylogenetic tree of two plants’ *UGT* proteins. The different colored arcs represent the UGT protein subfamilies. The tree was built using 156 PbUGTs from *Phoebe bournei*, 122 AtUGTs from *Arabidopsis thaliana*.

**Figure 4 plants-13-00722-f004:**
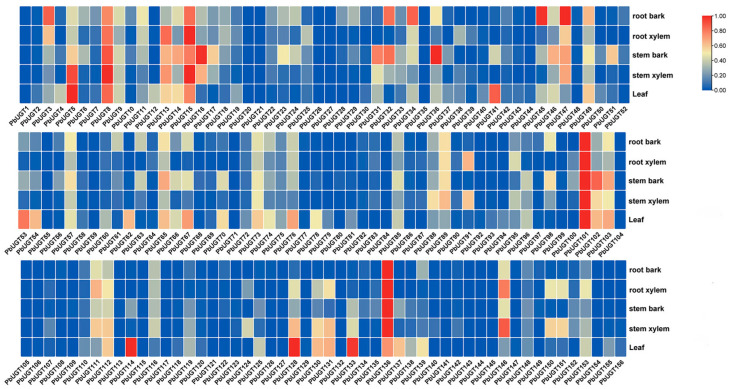
Expression heatmap of the *PbUGT* family in different organ and tissues. Differences in gene expression changes are shown by color, with warmer colors indicating higher expression.

**Figure 5 plants-13-00722-f005:**
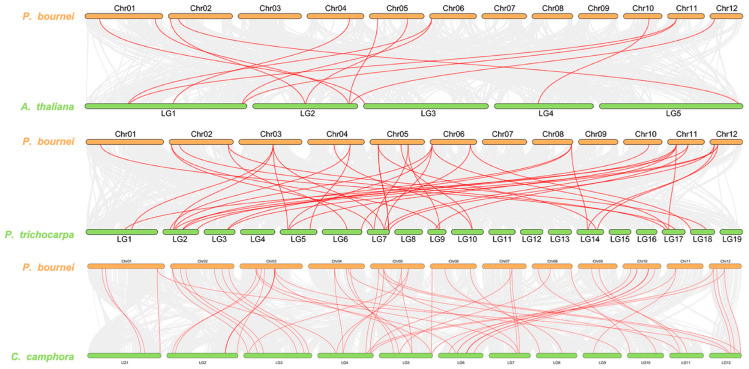
Synteny analysis of *UGT* genes among *Phoebe bournei*, *Arabidopsis thaliana*, *Populus trichocarpa*, and *Cinnamomum camphora*. The two rings in the middle represent the gene density of each chromosome, the gray lines represent the collinearity, and the red lines represent the highlighted collinearity of *UGT* genes.

**Figure 6 plants-13-00722-f006:**
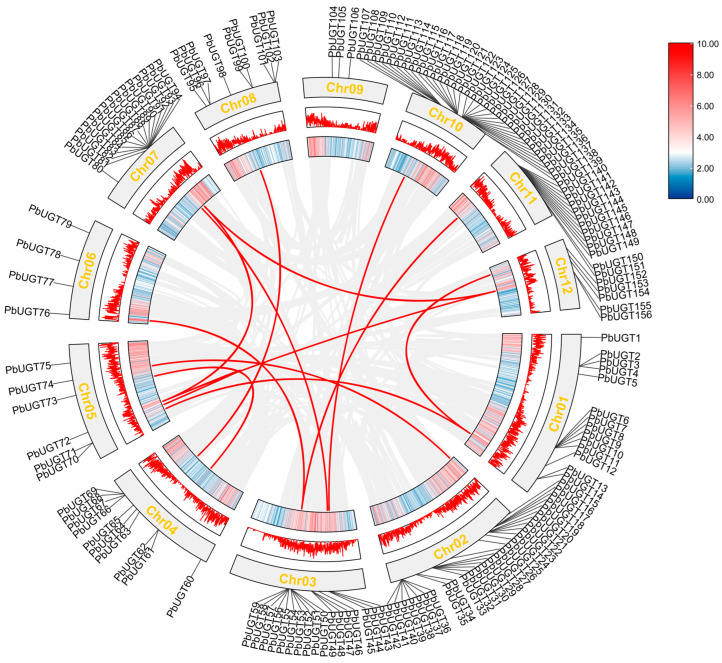
Distribution and collinearity analysis of *PbUGT* genes. The two rings in the middle represent the gene density of each chromosome, the gray lines represent the collinearity, and the red lines represent the highlighted collinearity of *UGT* genes.

**Figure 7 plants-13-00722-f007:**
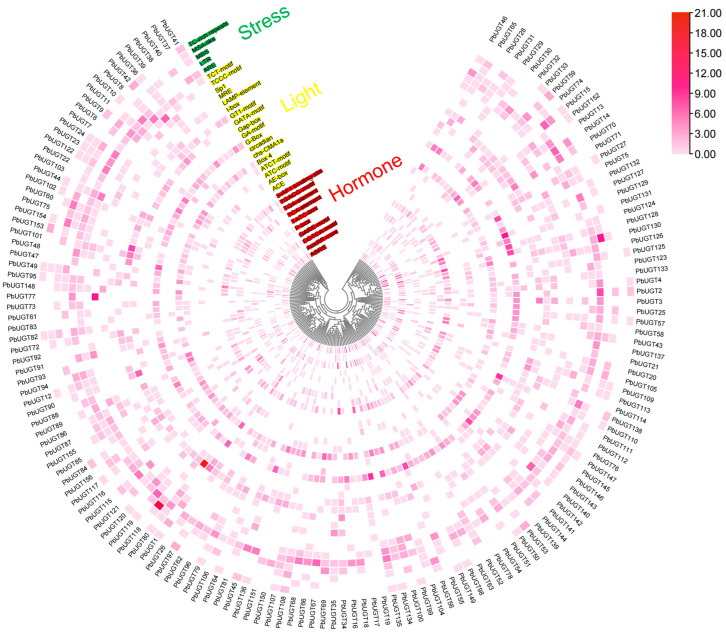
Enrichment analysis of stress-, hormone-, and light-responsive cis-elements in the promoter regions of the *PbUGT* gene family. Differences in the number of cis-elements are indicated by color, with darker colors indicating more cis-elements.

**Figure 8 plants-13-00722-f008:**
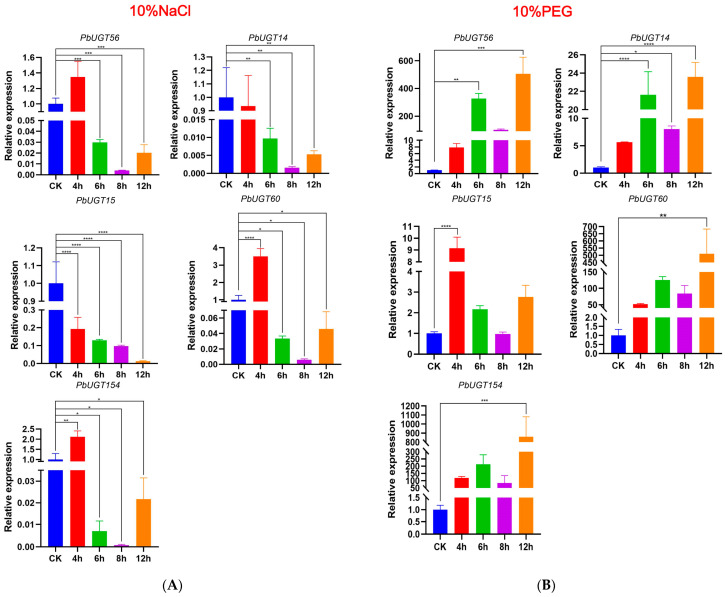
Expression profile of *PbUGT* genes responding to salt and drought stresses tested using RT-qPCR. (**A**) The relative gene expression levels under salt (10% NaCl) treatments for the same periods (0, 4, 6, 8,12 h). Control seedlings were treated with distilled water. (**B**) The relative gene expression levels under drought (10% PEG) treatments for the same periods (0, 4, 6, 8,12 h). Control seedlings were treated with distilled water. (* *p* < 0.05, ** *p* < 0.01, *** *p* < 0.0005, **** *p* < 0.0001).

**Table 1 plants-13-00722-t001:** Physical and chemical properties of *PbUGT* family members.

Name	Average Molecular Weight	Theoretical pI	Asp + Glu	Arg + Lys	Instability Index	Aliphatic Index	GRAVY
Distribution range	14,948.04–113,398.82	4.62–9.8	17–127	11–110	32.1–72.59	71.5–102.3	−0.467–0.273
Average value	50,994.75	5.87	54	44	44.31	88.18	−0.121
Minimum value member	*PbUGT*66	*PbUGT*140	*PbUGT*154	*PbUGT*154	*PbUGT*144	*PbUGT*136	*PbUGT*66
Maximum value member	*PbUGT*89	*PbUGT*1	*PbUGT*89	*PbUGT*9	*PbUGT*122	*PbUGT*75	*PbUGT*154

**Table 2 plants-13-00722-t002:** Primer sequences of quantitative RT-qPCR.

Primer	Sequence (5′~3′)
*PbUGT14*(*OF21457-F*)	TGCATGGGGCCTCAAAGG
*PbUGT14*(*OF21457-R*)	GGGCGAGCACTTCCAACT
*PbUGT15*(*OF04928-F*)	GTTTCCGCAGTGGGGTGA
*PbUGT15*(*OF04928-R*)	CGGTCCCCCAACAACCTC
*PbUGT56*(*OF23900-F*)	GTTCGTGGGGTGGAGTGG
*PbUGT56*(*OF23900-R*)	GGGTGGGCCAGAACTTCC
*PbUGT60*(*OF22991-F*)	TCGCCGCATCGAATCCTC
*PbUGT60*(*OF22991-R*)	AGGCGAGGGATTCAGGGT
*PbUGT154*(*OF17153-F*)	GCGGGCAGAGGTTCTTGT
*PbUGT154*(*OF17153-R*)	TTGCGACGCCCATGAGTT
Internal reference gene (*EF1α-F*)	CATTCAAGTATGCGTGGGT
Internal reference gene (*EF1α-R*)	ACGGTGACCAGGAGCA

## Data Availability

The whole-genome data and annotation files of *Phoebe bournei* were acquired from the China National GeneBank DataBase (CNGBdb) with lookup number CNP0002030 (https://db.cngb.org/ (accessed on 1 January 2023)).

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
