# Peer review of "Analysis of the UDP-Glucosyltransferase (UGT) Gene Family and Its Functional Involvement in Drought and Salt Stress Tolerance in Phoebe bournei"

_plants, 2024, doi:10.3390/plants13050722_

Round 1
Reviewer 1 Report
Comments and Suggestions for Authors
Main comments:
The authors undertook innovative comprehensive studies of gene structures and chromosome locations of very interesting unique endangered plant that is Phoebe bournei. The manuscript is generally written correctly, in accordance with the guidelines for scientific works. The information in the 'Materials and Methods' section should be completed.
Detailed comments and suggestions:
Abstract:
It is necessary to add how the stress was obtained: drought and salt stress.
Introduction:
The stresses used in the research should be characterized, what research has already been carried out with these stresses and what are the characteristics of plants' tolerance to the described stresses.
Results
Line 272-273: Figure 8. Why is 1.9 M NaCl given at the bottom of the graph and 10% NaCl at the top?
Materials and Methods
Divide the chapter into subchapters (subtitles)
Line 412-419: You should describe in detail what artificial growth conditions were: day/night temperature, day length, light intensity, type of lighting (LED, fluorescent, etc.). What the plants grew in (pots, containers, their capacity, type of soil, substrate, growing medium).
How were compounds causing osmotic stress added to the medium? Was it water culture? There is no description of how stress-inducing substances were given. Why was this concentration of PEG and NaCl chosen (based on what, what research?)
Line 418: In the graph titles (Fig. 8) it is written 10% NaCl and in 'Materials and methods' (1.9 M NaCl solution). Why is it not also given for NaCl concentration of 10% as for PEG?
Line 420: Which leaves were collected for subsequent RNA extraction?
Comments on the Quality of English Language
Minor stylistic errors in sentence structure.
Author Response
We feel great thanks for your professional review work on our manuscript. As you are concerned, there are several problems that need to be addressed. According to your nice suggestions, we have made corrections to our previous manuscripts, the detailed corrections are listed below.

Reviewer 2 Report
Comments and Suggestions for Authors
The manuscript by Guan and co-authors provides a valuable contribution to understanding Uridine diphosphate glycosyltransferases, specifically focusing on UGT genes in Phoebe bournei. The authors conducted a systematic analysis, identifying and categorizing the PbUGT genes in the P. bournei genome, and investigated their expression patterns across tissues, evolutionary context and responses to drought and salinity.
The manuscript is very well-written and provides valuable insights into the functional aspects of PbUGT genes, but there are several critical remarks that can be made:
1. The abstract lacks a clear contextualization of why studying UGT genes in Phoebe bournei is important. Readers may be left wondering about the broader implications of this research. Please consider adding some sentences to better engage the audience and articulate the significance of the study.
2. Several aspects are redundantly emphasized in the abstract, particularly the repeated mention of the identification of 156 PbUGT genes. Please rephrase these two sentences to avoid redundancy: “Therefore, we conducted a systematic analysis to identify 156 PbUGT genes and provide a comprehensive examination of this family. We identified 156 UGT genes within the entire P. bournei genome, all of which contained the PSPG box”.
3. Please arrange the keywords in alphabetical order, avoiding repetition of words present in the manuscript title.
4. Lines 101-105: The authors present average values/length/weight. The rationale behind providing these averages and their biological significance remain unclear. It would be beneficial to elaborate on their biological implications.
5. Lines 138-140: There is mentioning about the identification of 10 motifs and a range of motifs in PbUGT family members from three to 13. Clarification is needed regarding how the range extends from three to 13 when 10 motifs are identified.
6. In the present version of the manuscript, the quality of Figures 2-4 and Figure 7 is very low, making it impossible to distinguish different components and their labels. Please replace with higher quality figures.
7. Consider adding a brief section in the Conclusions that discusses potential future directions for research. Identifying unanswered questions or highlighting areas that merit further investigation could assist researchers interested in expanding upon the current study.
Minor issues:
- Line 13: Replace "which areregulated" with "which are regulated".
- Lines 168, 171, 203, 206, 207, 209, 216, 217, 384-387, 389, 391: Italicize Latin names of the plant species.
- Include clear subheadings in the Materials and Methods section. Currently, it is challenging to discern the beginning and end of the description for each technique and procedure.
Comments on the Quality of English LanguageThe English language of the manuscript is fine.
Author Response

(The authors gave the same response as above.)

Reviewer 3 Report
Comments and Suggestions for Authors
Title: Analysis of UGT gene family and functional involvement in drought and salt stress tolerance in Phoebe bournei
Abstract: Concise and representative of the findings
Line 13, Uridine diphosphate glycosyltransferases (UDP-GTs, UGTs), which are regulated
Line 14, In vivo, the activity of UGT genes can affect
Line 19-21, Therefore, we conducted a systematic analysis and identified 156 UGT genes within the entire P. bournei genome, all of which contained the PSPG box
Introduction:
Line 35-36, and solubility of receptors [1].
Line 65, while Zea mays
Line 75, economic value
Results:
This section is well detailed. Scientific names of the organisms must be corrected to italics form (Example lines 168, 171, 203-209). Overall, Except figure 8, other figures are overcrowded and clarity needs improvement.
Line 98, within the entirety of the genome of P. bournei
Line 99, Table 1 is not appropriately included in the text
Line 105, range from 4.62 to 9.80
Line 133, Figure 1, include the picture with more clarity
Line 138, in P. bournei
Discussion: Discussed the results appropriately. Scientific names of the organisms must be corrected to italics form.
Line 311, sylation in vivo
Materials and methods: This section needs subtitles
Author Response

(The authors gave the same response as above.)

Reviewer 4 Report
Comments and Suggestions for Authors
Dear Authors!
In the analysed work, a family of UGT genes has been identified in P. bournei that encode enzymes involved in the glycosylation of various substrates and are thus important in plant growth and development. The family consists of 156 PbUGT genes. By applying all possible available bioinformatic methods, a comprehensive characterisation of this gene family is given. Based on evolutionary analysis, 14 subfamilies of UGT genes were identified. The exon-intron structure of the genes, domain structure and various characteristics of the proteins encoded by these genes have been studied. A comparative study of the genes of this family in some plant species was carried out. The cis-elements in the promoter zone of the genes were studied.
Undoubtedly, interesting and useful information was obtained. Now there are many such articles. They expand the knowledge about this or that group of genes, but most often it concerns one or two previously unstudied plant species. Thus, original information is obtained only for one species and, nevertheless, such information is useful.
1. The aim of the study was to elucidate the mechanisms of PbUGT gene regulation under drought and salt stress conditions. The authors did not fulfil this task. They studied the transcript levels of 5 genes from the whole family under 1.9 M NaCl and 10% PEG6000.
2- The selection of genes for analysis is also questionable. Genes that had a higher number of cis-elements, presumably involved in the response to some stressors, were taken. However, there is no direct correlation between the number of cis-elements and the pattern of gene expression. Identical cis-elements under different conditions and in different positions may show different functional activity.
3. A definite regulation of the level of transcripts of the studied genes under the conditions of action of the studied factors was obtained. The authors concluded from these data that these genes are important for overcoming stress by the plant. In fact, the conclusion is overly optimistic. The results presented suggest an uncertain assumption that these genes may be somehow involved in plant stress tolerance.
4. Judging from the methodical part and the analysis of the results obtained, the authors do not have sufficient experience to study the mechanisms of plant resistance.
5. Salt stress. Why NaCl concentration of 1.9 M was taken. For the first time in my practice such a high concentration was encountered. I request the authors to provide a concentration curve of the choice of NaCl concentration. It would be useful to see if the seedlings will survive after exposure to such concentration? If all of them die, there is no point in such work. It would be a good idea to take photographs of the seedlings under study. The authors should describe in detail the conditions of plant cultivation (temperature, photoperiod, humidity), the method of cultivation and treatment of plants with both NaCl and 10% PEG6000. Such information is absolutely necessary to assess the validity of the experimental setup.
6. 10% PEG6000. In this paper, the authors did not study drought as such, but PEG-induced osmotic stress, one of the negative consequences of which is water deficit. For this reason, it would be correct to abandon the term "drought" in the manuscript, and to call the negative factor acting on plants as water deficit, which must be characterised in a mandatory manner. It is necessary to characterise tissue water content, relative water content, and osmotic and/or water potential. Without such data, there is no point in talking about the effect of PEG6000 on the plant and gene expression.
After addressing the noted deficiencies, the manuscript may be reconsidered if the editor deems it feasible. The second option may be to remove all resistance data, but this would be a different paper that the journal may not accept for a different reason.
Author Response

(The authors gave the same response as above.)

Round 2
Reviewer 1 Report
Comments and Suggestions for Authors
The authors of the manuscript tried to correct the content, but they did it partially and incorrectly in many places.
Introduction: The authors did not improve the text, the first review included a note: "What research (publications) has already been carried out with these stresses and what are the characteristics of plants' tolerance to the described stresses" What concentrations caused the stresses?
Line 25-26: The phrase needs to be corrected “Additionally, We will stress the seedlings with 10% NaCl and 10% PEG-6000”
Line 103-111. The inserted text is incorrect and repeats earlier text in this chapter.
Line 469: What was the capacity of the pots?
Line 471: Temperature at what degrees?
Line 472: How was light intensity measured? Why was the intensity so low?
Line 476-478: The authors did not respond to the first review and did not state how compounds causing osmotic stress were added to the substrate? What was it done in? Why was this concentration of PEG and NaCl chosen (based on what, what research?)
Comments on the Quality of English LanguageWriting style as well as specialized vocabulary requires improvement.
Author Response
We tried our best to improve the manuscript, and made changes marked in review mode in the revised paper that did not affect the content and framework of the paper. We sincerely thank the reviewer for their enthusiastic work and hope that the correction will be approved.
Thank you very much again for your comments and suggestions.

Reviewer 4 Report
Comments and Suggestions for Authors
Dear Editor
I think the article got better after the correction.
However, the authors do not distinguish between an organ and a tissue. They refer to any organ of a plant as a tissue. As we know, an organ is usually made up of several or many tissues. This is incorrect and needs to be corrected (lines 211-228). Figure 4
After this minor edit, the paper can be accepted for publication in the journal.
All the best
Author Response
We tried our best to improve the manuscript, and made changes marked in review mode in the revised paper that did not affect the content and framework of the paper. We sincerely thank the reviewer for their enthusiastic work and hope that the correction will be approved.Thank you very much again for your comments and suggestions.

Round 3
Reviewer 1 Report
Comments and Suggestions for Authors
The authors of the manuscript have partially corrected the text, but it is not yet fully correct
Line 25: The correct form for a summary in English is still not used. Overall, the summary is also not written in a good style.
Line 469: pots capacity should be given in SI units?
Line 476-478: The authors did not write in the manuscript for the first and second reviews how exactly they introduced stressful solutions for the plants, watered the pots with these solutions, or how they removed the plants from the pots and placed them in beakers? Very short, superficial description: “Drought stress was induced using a 10% PEG-6000 solution”
Comments on the Quality of English LanguageMinor stylistic errors in sentence structure.
Author Response
We tried our best to improve the manuscript, and made changes marked in review mode in the revised paper that did not affect the content and framework of the paper. We sincerely thank the reviewer for their enthusiastic work and hope that the correction will be approved.
